# Contractile and Structural Properties of Detrusor from Children with Neurogenic Lower Urinary Tract Dysfunction

**DOI:** 10.3390/biology10090863

**Published:** 2021-09-03

**Authors:** Navroop Johal, Kevin X. Cao, Boyu Xie, Michael Millar, Reena Davda, Aamir Ahmed, Anthony J. Kanai, Dan N. Wood, Rita I. Jabr, Christopher H. Fry

**Affiliations:** 1Department of Urology, Great Ormond St Hospital for Children NHS Foundation Trust, London WC1N 3JH, UK; regnnsj@icloud.com (N.J.); kevxcao@gmail.com (K.X.C.); 2School of Physiology, Pharmacology and Neuroscience, University of Bristol, Bristol BS8 1TD, UK; boyu.xie@bristol.ac.uk (B.X.); rita.jabr@surrey.ac.uk (R.I.J.); 3Queen’s Medical Research Institute, University of Edinburgh, Edinburgh EH16 4TJ, UK; Mike.Millar@ed.ac.uk; 4Departments of Oncology and Urology, University College London Hospital, London W1G 8PH, UK; reena.davda@nhs.net (R.D.); isovialdan@gmail.com (D.N.W.); 5Centre for Stem Cell Regeneration, King’s College London, London WC2R 2LS, UK; aamir.ahmed@kings.ac.uk; 6Department of Medicine, University of Pittsburgh, Pittsburgh, PA 15260, USA; ajk5@pitt.edu

**Keywords:** bladder, paediatric, neuropathic bladder, detrusor smooth muscle, contraction, immunohistochemistry

## Abstract

**Simple Summary:**

Disorders of bladder function can result from congenital spinal cord developmental defects and can remain in a significant number of patients despite surgical improvements to repair the primary defect. We studied the ability of bladder wall muscle from such patients to contract, a function essential to void collected urine and avoid urinary tract infections and potential damage to the kidneys. Tissue was taken when patients were several years old, at the time of surgical operations to improve bladder function. This tissue would otherwise have been discarded and was collected with the full ethical approval and consent of parents or guardians. We found that the ability of the bladder wall samples to contract was impaired and was generally stiffer; both of which would make it more difficult for the bladder to void urine. These functional changes were associated with a replacement of muscle with connective tissue (fibrosis). The experiments provide a pathway to devise strategies that might improve bladder function in these patients through reversal of the intrinsic tissue pathways that increase fibrosis.

**Abstract:**

Neurogenic lower urinary tract (NLUT) dysfunction in paediatric patients can arise after congenital or acquired conditions that affect bladder innervation. With some patients, urinary tract dysfunction remains and is more difficult to treat without understanding the pathophysiology. We measured in vitro detrusor smooth muscle function of samples from such bladders and any association with altered Wnt-signalling pathways that contribute to both foetal development and connective tissue deposition. A comparator group was tissue from children with normally functioning bladders. Nerve-mediated and agonist-induced contractile responses and passive stiffness were measured. Histology measured smooth muscle and connective tissue proportions, and multiplex immunohistochemistry recorded expression of protein targets associated with Wnt-signalling pathways. Detrusor from the NLUT group had reduced contractility and greater stiffness, associated with increased connective tissue content. Immunohistochemistry showed no major changes to Wnt-signalling components except down-regulation of c-Myc, a multifunctional regulator of gene transcription. NLUT is a diverse term for several diagnoses that disrupt bladder innervation. While we cannot speculate about the reasons for these pathophysiological changes, their recognition should guide research to understand their ultimate causes and develop strategies to attenuate and even reverse them. The role of changes to the Wnt-signalling pathways was minor.

## 1. Introduction

Neurogenic lower urinary tract (LUT) dysfunction results from either a congenital or acquired condition that affects lower urinary tract innervation. Congenital lesions include neural tube defects such as myelomeningocele, tethering of the spinal cord, and sacral agenesis [1,2]. Acquired lesions may occur from traumatic, infective, vascular or space-occupying causes. The majority of neuropathic bladders in the paediatric population are associated with myelomeningocele, the prevalence of which remains high despite a greater understanding of causative factors [3]. A current ‘two-hit’ hypothesis [4] is that a spinal defect exposes regions of the cord to amniotic fluid, which are subsequently damaged by fluid exposure, trauma or raised fluid pressures, leaving a motor neurone lesion that affects the lower urinary tract. Management is independent of the underlying condition and aims to allow or replicate normal bladder function, including urinary continence, as well as preservation of the upper urinary tract. Lower urinary tract (LUT) dysfunction may take several forms: detrusor overactivity with either sphincter over-/under-activity, or detrusor underactivity with similar variability of sphincter function [5] and reduced sensation on filling [6]. Furthermore, upper urinary tract damage can result from LUT dysfunction and, in some cases, a decrease in bladder compliance [6,7]; incontinence is usual [8]. Surgical management has greatly improved and resulted in better renal outcomes [9]. However, urinary tract problems persist and are difficult to treat without a clear understanding of the pathophysiology. In utero surgical repair has led to improvements in lower limb development but has yet to show changes to urological outcomes [10].

Apart from re-modelling of LUT nervous control, fundamental changes to the physiological properties of the bladder itself are associated with neurogenic LUT dysfunction in adults. This includes development of fibrosis in adult and paediatric human tissue [11,12] and changes to detrusor contractile function in adult animal models and adult human tissue [13,14]. However, to our knowledge there are no studies detailing the contractile and biomechanical properties of human detrusor from the paediatric population with congenital neuropathic bladders, and this was one objective of the study.

Crucial cellular pathways in the development of fibrosis are activation of a TGF-β pathway and expression of enzymes regulating collagen turnover, such as matrix metalloproteinases (MMPs) and their tissue inhibitors (TIMP) [15], supplemented by transformation of fibroblasts and epithelial cells to collagen-producing myofibroblasts [16]. These processes are augmented by the release of *Wnt*-ligand proteins mediated by transcription factors such as β-catenin, c-Myc and cyclin-D1. Modulation of *Wnt*-ligand regulation occurs in another congenital human bladder anomaly, exstrophy, by identification of coding changes [17] and immunohistochemical mapping of transcription factors [18]. A further objective of this study was to identify the potential role of *Wnt*-ligand mediated pathways in the development of fibrosis in the paediatric neurogenic neuropathic bladder.

## 2. Methods

### 2.1. Tissue Samples, Ethics and Preparations

A single biopsy sample per bladder was taken (up to 5 mm^3^) from two groups of patients. One group comprised patients with neurogenic LUT dysfunction (termed neuropathic), as assessed by filling urodynamics, due to congenital spinal cord anomalies and undergoing bladder augmentation (*n* = 14:4 males, 10 females; median age 65 [interquartile range (IQ) 60, 92] months, range 30–120 months). A comparator (control) group had normal bladder function as assessed by history and non-invasive urodynamic investigations (*n* = 14:10 males; 4 females; median age [interquartiles] 26 [IQ 18, 48] months, range 12–120 months). They were undergoing surgery for ureteric re-implantation or excision of urachal cysts. The ages of the two groups were significantly different (*p* < 0.01, Wilcoxon rank-sum test). Biopsy samples were taken from the anterior bladder wall, placed immediately in Ca^2+^-free Tyrode’s solution and used within 15–30 min. Mucosa was removed from the biopsy after blunt dissection, and remaining tissue was divided into pieces for (1) in vitro functional experiments and (2) fixation in 4% paraformaldehyde and storage in phosphate-buffered saline at 4 °C for histology and immunohistochemistry.

### 2.2. Solutions

Tyrode’s solution contained (mM): NaCl, 118; NaHCO_3_, 24; KCl, 4.0; MgCl_2_, 1.0; NaH_2_PO_4_, 0.4; CaCl_2_, 1.8; glucose, 6.1; Na pyruvate, 5.0; gassed with 5%CO_2_:95% O_2_, for pH 7.4 at 36 ± 0.5 °C. Ca^2+^-free solution was (mM): NaCl, 132; KCl, 4.0; NaH_2_PO_4_, 0.4; MgCl_2_, 1.0; glucose, 6.1; Na pyruvate, 5.0; HEPES, 10.0, pH 7.4 with 1 M NaOH. The muscarinic receptor agonist carbachol and antagonist atropine as well as the purinergic receptor (P2X_1_) agonist α, β methylene ATP (ABMA) were diluted from aqueous 1 or 10 mM stock solutions. All chemicals were purchased from Sigma-Aldrich UK unless otherwise stated.

### 2.3. Active Tension Recording

Detrusor strips (1–2 mm diam; 4–5 mm length) were tied between an isometric force transducer and a fixed hook in a horizontal superfusion trough. Nerve-mediated contractions were generated by electrical field stimulation (EFS: 3 s trains, width 0.1 ms, frequency 1–40 Hz) and abolished by 1 µM tetrodotoxin. Contractions by direct muscle stimulation were generated by carbachol (0.1–30 µM) or ABMA (10 µM). Tension amplitude data were normalised to preparation cross-sectional area, as calculated from measurement of the length and diameter of the preparation in the superfusion trough.

### 2.4. Histology

Tissue was serially dehydrated in ethanol, cleared in xylene and paraffin-mounted. Three or more sections (5 µm) were mounted on TESPA-coated glass slides and stained with Verhoeff van Gieson (muscle yellow/orange; collagen, red). Whole-section images were acquired (Zeiss Axioscan Z1, Zeiss, Cambridge, UK), filtered digitally for noise on ImageJ, and amounts of muscle and connective tissue (collagen and elastin) measured using colour thresholding. Three random separate regions (50 × 50 µm) per section were measured, distant from any mucosa or obvious areas absent of tissue, and the average was recorded.

### 2.5. Biomechanical Measurements

Detrusor stress–strain relationships were obtained from preparations of control bladders (*n* = 8) and those with neurogenic LUT dysfunction (*n* = 7). Preparations were mounted in a similar arrangement as for active tension recording, except that a rod within a solenoid coil replaced the fixed hook. The rod could be rotated on application of a potential difference applied to the solenoid. Application of a 60 s square-wave potential difference rapidly stretched the preparation to generate a rapid increase in passive tension that partially relaxed to a new steady-state level, *T* (see Figure 2C). *T* was the estimated asymptote on fitting the relaxation phase to a single declining exponential function. The elastic modulus, *E*, is a measure of tissue stiffness and was calculated from the relationship: *E* = *T*/(∆*L*/*L*), where T has units of mN.mm^−2^ (kPa) in response to a fractional length change, strain (∆*L*/*L*; *L*, resting muscle length). ∆*L*/*L* varied between 0.06 and 0.28 to avoid excessive stretch when the stress–strain relationship becomes non-linear.

### 2.6. Multiplex Immunofluorescence Labelling and Quantitative Image Intensity Analyses

Tissue arrays were each made from 25 to 30 collections of paraffin-embedded tissue cores (1 mm diameter) and sections (5 µM) cut for antibody labelling. A BondmaX™ automated system (Leica BioSystems, Seoul, Korea) was an automated staining platform [18,19], and antibodies were optimised for concentration, pH-dependence and antigen retrieval. Those used, as per manufacturer’s instructions, were: β-catenin (ab22656, Abcam, Cambridge, UK), c-Myc (Novacastra/Leica Biosystems), cyclin-D1 (sc 718, Santa Cruz, Santa Cruz, CA, USA) and MMP-7 (ab4044, Abcam). The researcher was blinded to the tissue array arrangement. Fluorescent labels were used: FITC (488/517 nm, green, β-catenin), Cy3.5 (561/617 nm, red, c-Myc), Cy5 (633/671 nm, purple, cyclin-D1) and Cy3 (514/565 nm, yellow, MMP-7). For expression analyses, each section was imaged with a TCS SP8 confocal system (Leica) at 63x magnification. Then, from five regions under a further 6x times digital magnification, the number of pixels at each fluorophore channel was measured using a macro compiled within Huygens Professional software (SVI, Hilversum, The Netherlands) to measure individual channel intensities [18].

### 2.7. Data Presentation and Analysis

Data are mean ± SEM, *n* = number of biopsies. Significance between multiple data sets used ANOVA, followed by parametric post hoc tests; the null hypothesis was rejected at *p* < 0.05. Concentration–response or force–frequency curves were fitted to: *T* = (*T*_max_.*x*^k^)/(*x*^k^ + *k_m_*^k^), where *T*_max_ is maximum response at high stimulation frequency (*f*) or agonist concentration (*S*); *x* is different values of *f* or *S*; *k*_m_ is the value of *x* required to achieve *T*_max_/2; k is a constant.

## 3. Results

### 3.1. Contractile Properties

The contractile responses of isolated detrusor preparations from the neuropathic and control groups to nerve-mediated and agonist-induced stimulation were recorded. Absolute nerve-mediated contractions to EFS were significantly smaller at all frequencies than neuropathic compared to control (Figure 1A) with estimated *T*_max,EFS_ values of 2.10 ± 0.62 vs. 7.68 ± 2.50 mN.mm^−2^ (*n* = 10,14; *p* < 0.05). Furthermore, with four samples from the neuropathic group no EFS responses could be recorded, although they responded to contractile agonists. The frequencies required to reach *T*_max,EFS_/2 (*f*_1/2_) were not statistically different (11.1 ± 0.9 vs. 12.7 ± 1.9 Hz; *n* = 10,14; *p* = 0.330).

The responses to the contractile agonists carbachol and α, β-methylene ABMA were also significantly smaller in samples from the neuropathic group (Figure 1B). Maximum responses to carbachol from concentration response curves, *T*_max,carb_, were 8.76 ± 2.50 vs. 23.78 ± 5.19 mN.mm^−2^ (*n* = 14,13; *p* < 0.05); responses to 10 µM ABMA were 11.06 ± 2.50 vs. 1.60 ± 0.52 mN.mm^−2^ (*n* = 14,11; *p* < 0.05). In addition, the proportional reductions in tension to EFS, carbachol and ABMA were significantly different. The ratio of *T*_max,EFS_/*T*_max,carb_ was significantly less in bladder samples from the neuropathic group (0.24 ± 0.05 vs. 0.50 ± 0.11; *n* = 10,13; *p* < 0.01), interpreted as evidence of functional denervation (see Discussion). In addition, the contractile response to ABMA was less by a significantly greater extent than to carbachol (T_ABMA_/T_carb,max_: 0.24 ± 0.05 vs. 0.43 ± 0.05, *n* = 14,13, *p* < 0.05).

Despite the smaller contractile response to carbachol in the neuropathic group, there was greater potency of the agonist, as evident from an increase in p*EC*_50_ (−log_10_*EC*_50_: 6.25 ± 0.08 vs. 5.82 ± 0.07; *n* = 14,13; *p* < 0.001—Figure 1C). Atropine resistance was present to a similar extent in both groups (25.2 ± 4.7 vs. 27.1 ± 3.7% of contractions generated at 8 Hz EFS; *n* = 10,12; *p* = 0.813, neuropathic vs. control).

### 3.2. Histological Properties and Detrusor Stiffness

The smooth muscle and connective tissue percentages of sections from tissue from the neuropathic and control groups, stained with Verhoeff van Gieson mixture, were measured. Together, these comprised the bulk of the tissue. The smooth musle content of neuropathic tissue sections was significantly less than that of control bladders (52.7 ± 3.0 vs. 67.0 ± 3.0%; *n* = 9,9; *p* < 0.01, neurogenic LUT vs. control), with a corresponding increase in connective tissue (47.4 ± 3.0 vs. 33.0 ± 3.0, Figure 2A,B). The stiffness of the same bladder wall preparations used for histological obervations and measurement of contractile properties was measured as an estimate of elastic modulus, *E*. An example of an experiment (from a neuropathic group sample) is shown in Figure 2C. Shown are tension responses resulting from prolonged stretches (∆*L*) of 0.4, 0.6, 0.8, 1.0 and 1.2 mm of a 5 mm resting length (*L*) preparation. The asymptotic values of tension (*T*) were plotted as a function of ∆*L*/*L* to yield a linear plot with slope of the elastic modulus, *E*, as an index of passive stiffness. Values of *E* were significantly greater in preparations from the neuropathic group, compared to those from the control group (57.0 ± 10.9 vs. 26.4 ± 2.6 kPa, *n* = 7,8, *p* < 0.05; Figure 2C). However, one preparation from the neuropathic group had a very low stiffness (Figure 2C, right).

**Figure 2 biology-10-00863-f002:**
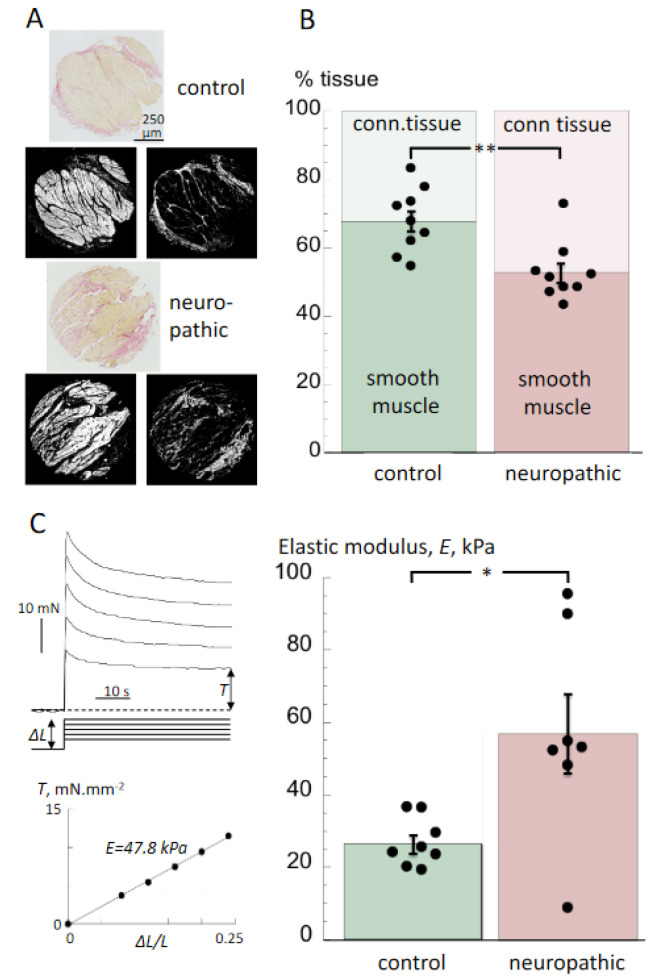
Histological and biomechanical properties of detrusor from neuropathic and control bladders (**A**): Verhoeff van Gieson stained sections of tissue from control (upper trio) and neuropathic (lower trio) bladders. Shown below each primary section (yellow/orange: smooth muscle: red; collagen) are grey scale images of the yellow/orange (left) and red (right) areas obtained from ImageJ files. (**B**): Percentages of detrusor and connective tissue in samples from neuropathic and control bladders (mean ± SEM, *n* = 9,9 neuropathic vs. control; ** *p* < 0.01). Dot plots show the percentages of smooth muscle (**C**): Sample traces (left) of tension values upon rapid, sustained stretches for estimation of the elastic modulus. The asymptotic tension values, *T*, were recorded for a series of sustained strain changes, ranging from ∆*L* = 0.08–0.24 in equal increments. The plot below shows the relationship between *T* and ∆*L*/*L* with the slope equal to the elastic constant, *E* (see Methods for more details). Bar chart with dot plots of values of *E* from neuropathic and control bladders (mean ± SEM, *n* = 7,8); * *p* < 0.05 neuropathic vs. control.

### 3.3. Multiplex Immunofluorescence Labelling

Expression of key proteins in cell turnover and fibrosis was measured from high-power immunofluorescent images (×20, upper and ×63, lower; Figure 3A) of control and neuropathic bladders; three sections from each sample were imaged. DAPI counts, as a nuclear stain and indicative of cell numbers, were not different between the two groups (9.73 ± 1.32 × 10^8^ vs. 9.79 ± 2.05 × 10^8^ pixels per section; *n* = 9,5; Figure 3B), indicating that any alterations to other labels were not due to different numbers of cells. MMP7 protein expression was not different in the two groups, nor was it for cyclin D1 or β-catenin; however, there were significantly (*p* < 0.01) fewer counts of c-Myc expression in samples from the neuropathic group.

## 4. Discussion

These in vitro experiments recorded quantitative physiological and structural differences between detrusor muscles from two paediatric patient groups: one with lower urinary tract dysfunction associated with congenital neurogenic disorders and one a comparator (or control) group with normal bladder function. The principal difference between them was the presence or absence of lower urinary tract dysfunction, as assessed by normal clinical assessment prior to surgery, and it is on this basis that data were compared. However, several caveats are relevant to most research with isolated human tissue. Firstly, the control group will have had urinary tract abnormalities requiring surgical intervention, but nevertheless had normal non-invasive urodynamic function. Secondly, patients with congenital neurogenic disorders will have received various forms of medical management in the interim between initial corrective surgery and later bladder augmentation. NLUT is a heterogeneous patient group with different degrees of severity in acquired effects that may further impact on bladder function, including episodes of acute retention, urinary tract infections and chronic kidney disease that all may contribute to detrusor changes. Thirdly, neuropathic patients requiring augmentation who formed the study group will not represent all such NLUT individuals, and so conclusions may only be pertinent to this sub-group. Finally, the two groups could not be age-matched precisely, otherwise the number of biopsy samples would have been limited severely. With respect to the latter point we consider this would underestimate differences between the two groups, as there is a natural increase in smooth muscle content and contractile function with age [20]; thus, the neuropathic disorder group was older but showed reduced values (see Methods). However, the study demonstrates that differences in lower urinary tract function between the two paediatric patient groups are associated with in vitro structural and functional differences.

### 4.1. Contractile Properties of Detrusor from Bladders with Neurogenic LUT (NLUT) Dysfunction

Nerve-mediated and agonist-induced contractions were significantly smaller with detrusor muscle in the NLUT group, compared to those from normally functioning bladders. The data suggest that several factors contribute to this reduced contractile performance. Firstly, there was greater fibrosis as shown by a reduced smooth muscle:connective tissue ratio. However, there was also evidence of functional denervation as nerve-mediated contractions were proportionately even smaller than agonist-induced contractions, as seen by the reduced *T*_max,EFS_/*T*_max,carb_ ratio. Finally, purinergic-mediated contractions, with the P2X_1_ receptor agonist ABMA, were reduced proportionately more than those to carbachol, i.e., the *T*_max,carb_/*T*_ABMA_ ratio was reduced. Because nerve-mediated ATP and ACh release both contributed to contractile development, as deduced from significant atropine resistance, decreased efficacy of ATP would also contribute to tension reduction in the NLUT group. These three factors, which would negatively impact on reduced contractile performance, would be partly offset by the increased potency of nerve-mediated Ach release, as seen by the increase in the carbachol pEC_50_.

Overall, this complex picture of contractile decline is different from that observed in human detrusor from other congenital anomalies such as bladder exstrophy or posterior urethral valves [18,19]. In these cases, replacement of detrusor muscle by connective tissue was also deemed to be a major factor but, crucially, functional denervation was not evident. The relatively smaller efficacy of the purinergic agonist on detrusor from the NLUT group was also more evident than with exstrophy or neuropathic detrusor.

Super-sensitivity of detrusor smooth muscle to muscarinic receptor agonists and its association with denervation has been proposed to contribute to development of detrusor overactivity in adults [13,21,22] and, if so, this would also be relevant to the NLUT group. However, this observation is not universal and was absent from detrusor from adult human patients with overactive bladders associated with outflow tract obstruction or idiopathic causes [14]. It should be noted that detrusor from functionally normal paediatric bladders is less sensitive to carbachol than equivalent adult tissue [14,18], and increases with development to adulthood [20]. The increase in potency measured in this cohort of NLUT patients means that this development may be accelerated, but the linkage to overactivity remains unproven.

### 4.2. Histological and Biomechanical Properties of Tissue from Bladders with Neurogenic LUT Dysfunction

Biopsy samples of the detrusor layer from this cohort showed an increased proportion of connective tissue compared to bladders with normal function. This was reflected in a greater passive stiffness of such samples as determined by measurement of the elastic constant, *E*. Connective tissue contains collagen fibrils, embedded in a ground substance, that have a greater intrinsic elastic modulus than muscle tissue [23,24]. This is consistent with poor compliance and upper tract damage in a large proportion of patients with myelomeningocele or sacral agenesis [25,26], as well as increased connective tissue deposition in animal models and humans with spinal cord injury, in particular deposition of the subtype of collagen (type-I) with greater intrinsic stiffness [27,28]. Medical management of neurogenic bladders with antimuscarinic agents or botulinum toxin can improve continence and renal outcomes by reducing detrusor overactivity and improving compliance. Their modes of action remain unclear, but two are suggestive: reduced afferent filling sensation, as both agents attenuate stretch-dependent ATP release from urothelium that activates afferent nerves [29,30], or an antifibrotic action as suggested for botulinum toxin [31]. Further studies introduced the caveat that botulinum toxin is less efficacious in children with substantial fibrosis and very low bladder compliance [27,32], suggesting that an antifibrotic action may be of less importance than hitherto proposed. Nonetheless, reversal or prevention of bladder fibrosis will be desirable as a strategy to protect the upper tract against excessive pressures and minimise loss of smooth muscle tissues. Several strategies have been suggested [33,34], and one, injection of the hormone relaxin, has proved successful in fibrosis induced by X-ray irradiation [35].

### 4.3. Molecular Pathways Underlying Fibrosis

Multiplex immunofluorescent labelling showed few changes to target proteins that have been implicated in the development of fibrosis in other conditions. The matrix metalloproteinase MMP-7 labelling was not significantly different; nor was the cell cycle regulator cyclin D1; nor β-catenin, an intermediate in the canonical Wnt-signalling pathway. The only protein studied that showed a change in neurogenic LUT dysfunction bladders was a reduction in c-Myc, a multifunctional nuclear regulator of gene transcription [36]. This was consistent with a reduction in co-localisation with the nuclear marker DAPI (Figure 3C). c-Myc regulates a large number of genes, and an increased expression in many tissues is associated with fibrosis [37,38], although reduced expression improves the function of many tissues [39]. Of significance in the context of fibrosis development in paediatric human bladders subjected to different congenital anomalies is that the pattern of protein expression varies greatly. Thus, in exstrophy tissue, MMP-7, β-catenin and c-Myc all showed reduced expression [18], whereas in detrusor from obstructed bladders MMP-7 and cyclin-D1 expression were increased, but β-catenin and c-Myc were reduced [19]. The major conclusion that may be drawn is that although tissues from all bladder anomalies developed fibrosis, the cellular pathways responsible were different. Thus, each condition requires separate study to understand their unique pathogenesis and origin of fibrosis.

## 5. Conclusions

Detrusor from this group of children with NLUT dysfunction was less contractile and was associated with loss of smooth muscle when compared to connective tissue. The remaining detrusor was, however, more sensitive to the muscarinic receptor agonist carbachol, although less responsive to the purinergic receptor agonist, ATP. The increase in connective tissue was positively associated with a greater stiffness of detrusor in the non-contractile state. The increase in connective tissue is not associated with dysregulation of the Wnt-signalling pathway, an important regulator of fibrosis, and this differs from poorly contractile detrusor from other congenital bladder anomalies. This implies that although fibrosis is a consistent feature of several bladder pathologies, the cellular pathways controlling it are different, so that each pathology deserves separate study.

## Figures and Tables

**Figure 1 biology-10-00863-f001:**
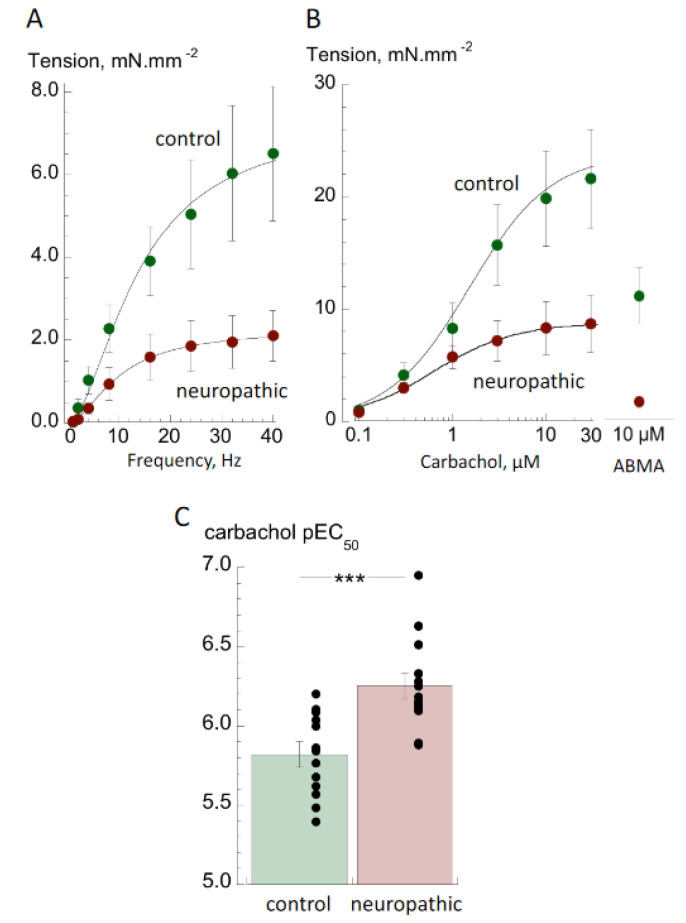
Contractile properties of detrusor preparations from neuropathic and control bladders. (**A**): Nerve-mediated contractions as a function of stimulation frequency (mean ± SEM, *n* = 10,14; neuropathic vs. control). (**B**): Carbachol concentration–response curves (*n* = 14,13) and the response to 10 µM ABMA (*n* = 14,11). (**C**): Dot plots and bar chart (mean ± SEM) of carbachol p*EC*_50_ values (*n* = 14,13), *** *p* < 0.001 neuropathic vs. control.

**Figure 3 biology-10-00863-f003:**
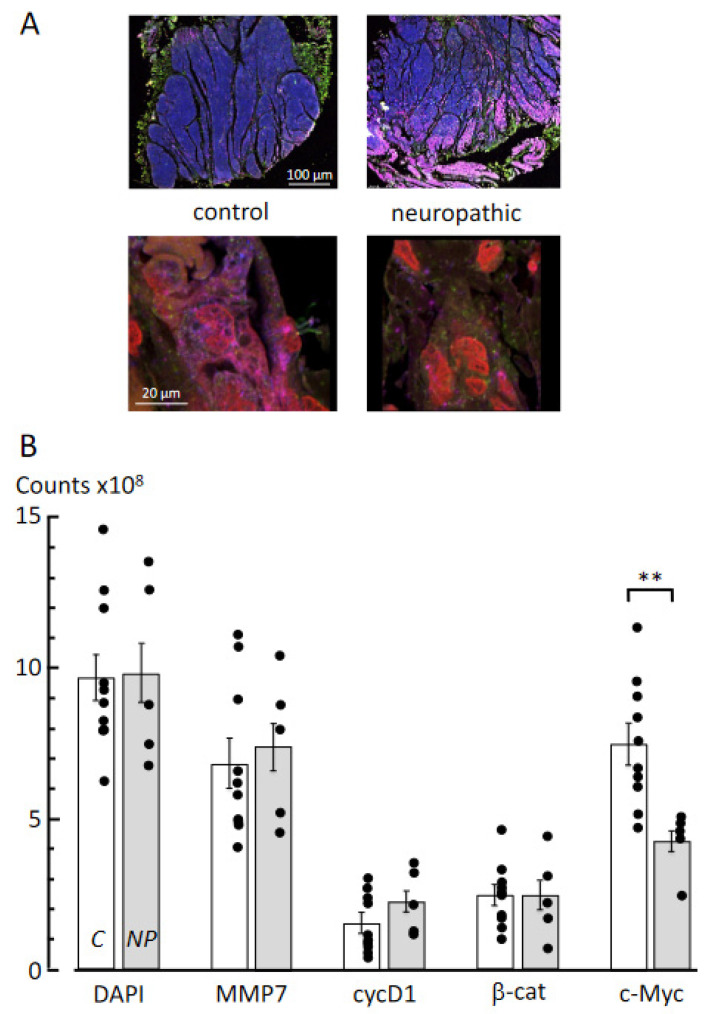
Multiplex immunofluorescence image analysis of control and neuropathic bladder samples. (**A**): Protein expression for MMP7, cyclinD1, β-catenin and c-Myc (DAPI used as counterstain) was measured in bladder tissue samples. Representative images at low magnification (20×, top lane, scale bar 50 µm) and high magnification (63×, bottom panel, scale bar 20 µm) for control and neuropathic bladder tissue. Images are composite overlays of four fluorophores: Cy3 (yellow) for MMP-7; Cy5 (purple) for cyclin-D1; FITC (green) for β-catenin; Cy3.5 (red) for c-myc; DAPI nuclear label (blue). The higher magnification images show each label clearly, relative to the low-magnification images. (**B**): Quantitative, intensity, analysis was performed on 63× images subsequent to fluorophore channel separation (see methods) from control (C, empty bars *n* = 9) and neurogenic LUT dysfunction bladder samples (NP, grey bars, *n* = 5), using Huygens analysis software (SVI). The expression of c-Myc was significantly (** *p* < 0.01) different between control and neurogenic LUT dysfunction bladders.

## Data Availability

The data that support the findings of this study are in the text of the paper and are available from the corresponding author upon reasonable request.

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
