# Peer review of "Contractile and Structural Properties of Detrusor from Children with Neurogenic Lower Urinary Tract Dysfunction"

_biology, 2021, doi:10.3390/biology10090863_

Round 1
Reviewer 1 Report
The condition of neuropathic bladder is complex and not fully comprehended. Changes in patterns of innervation and alterations in bladder histology have been widely documented in the adult tissue but not so much in the paediatric condition and neuropathic bladder is still irreversible, as no fully effective treatment exists.
In the manuscript authors analyse bladder biopsies from neurogenic bladder paediatric patients in terms of fibrotic tissue development and changes in contractile and biomechanical properties, as well as provide some mechanistical insights into the potential role of Wnt-ligand mediated pathways in the disease pathophysiology.
Despite good pieces of information on the human neuropathic bladder pathology presented by the authors, that are some matters that should be clarified and/or explained in more detail. Please find below my main concerns:
(Introduction, Line 41-43) Neuropathic bladder results from the damage of multiple nerves and not only caused by the lesion of motor neurons. Do foetal spinal defects exclusively expose ventral regions of the spinal cord to the amniotic fluid? For example, in the case of spinal cord traumas bladder overactivity is accepted to be a consequence of sensory afferents sprouting and uncontrolled activity by regulatory areas in the brain. Please clarify.
(Introduction, Lines 49-51) Damage in neuronal pathways controlling the urinary bladder are often irreversible and normal function is hardly restored. Might it be the case that surgical interventions are only performed when major structural and functional changes of the urinary tract had already occurred? Is there an optimal time in terms of foetal development to intervene to avoid such complications?
(Methods, Tissue samples, ethics and preparations) Voiding is a reflex activity in infants up to 24-36 months of age. The ability to voluntary void surges with maturation of the central nervous system. Whereas changes in bladder tissue and contractile activity occur during this developmental time is not addressed by the authors and should be taking into consideration. Please add some information and explain why such a range of ages 3-120 months was considered in the study.
(Methods, Tissue samples, ethics and preparations) Was the size of the biopsies controlled? The condition of neurogenic bladder is characterized by an increase in the size of the urinary bladder. Did author collect data on bladder sizes? Again, range of ages of the patients involved in this study is big and this translates into differences in whole tissue size, which might impact proteins quantification and comparisons among groups. Please clarify how this was taken into consideration when selecting the tissue for analysis and generating results.
(Results, Figure 2) Figure 2 should be shown in the manuscript after figure 1, please correct. Also, the legend should include what the yellow/orange and red staining represents in terms of connective tissue and smooth muscle.
(Results, Figure 3) Legend of figure 3 is incomplete. Immunofluorescence images should be accompanied by the description of the markers used. It is not clear how the quantification of the different proteins was performed.
(Results) In addition to the analysis of proteins associated with fibrosis, it would be relevant to study the innervation of the bladder in the paediatric tissue biopsies. To corroborate the findings in tissue tension/ contractility authors should consider conducing immunofluorescence protocols for the detection and quantification of cholinergic and purinergic markers.
Author Response
Referee 1
Thank you for your careful consideration of our manuscript.
Changes to the text to your particular questions are highlighted in the text as red.
The condition of neuropathic bladder is complex and not fully comprehended. Changes in patterns of innervation and alterations in bladder histology have been widely documented in the adult tissue but not so much in the paediatric condition and neuropathic bladder is still irreversible, as no fully effective treatment exists.
In the manuscript authors analyse bladder biopsies from neurogenic bladder paediatric patients in terms of fibrotic tissue development and changes in contractile and biomechanical properties, as well as provide some mechanistical insights into the potential role of Wnt-ligand mediated pathways in the disease pathophysiology.
Despite good pieces of information on the human neuropathic bladder pathology presented by the authors, that are some matters that should be clarified and/or explained in more detail. Please find below my main concerns:
1 (Introduction, Line 41-43). Neuropathic bladder results from the damage of multiple nerves and not only caused by the lesion of motor neurons. Do foetal spinal defects exclusively expose ventral regions of the spinal cord to the amniotic fluid? For example, in the case of spinal cord traumas bladder overactivity is accepted to be a consequence of sensory afferents sprouting and uncontrolled activity by regulatory areas in the brain. Please clarify.
The main congenital conditions that can cause neurogenic LUT dysfunction are listed in the Introduction (paragraph 1, lines 2-3). We had an inclusion criterion as those paediatric patients with neuropathic LUT dysfunction due to a congenital spinal cord defect (Methods, paragraph 1, lines 5-6) and did not distinguish between the different spinal cord lesions.
2 (Introduction, Lines 49-51). Damage in neuronal pathways controlling the urinary bladder are often irreversible and normal function is hardly restored. Might it be the case that surgical interventions are only performed when major structural and functional changes of the urinary tract had already occurred? Is there an optimal time in terms of foetal development to intervene to avoid such complications?
We appreciate your important point. Patients were undergoing surgical interventions when the bladder has become ‘hostile’ - e.g. when the kidneys were at risk of damage. Bladder augmentation was performed to reduce the risk of urine infections, the chance of renal impairment and improve urinary continence.
We appreciate that the patient population that we sampled may not represent all such patients with neurogenic LUT dysfunction. To that end we have included a new paragraph at the very beginning of the Discussion to put this study into better perspective.
3 (Methods, Tissue samples, ethics and preparations) Voiding is a reflex activity in infants up to 24-36 months of age. The ability to voluntary void surges with maturation of the central nervous system. Whereas changes in bladder tissue and contractile activity occur during this developmental time is not addressed by the authors and should be taking into consideration. Please add some information and explain why such a range of ages 3-120 months was considered in the study.
Surgeries were performed on all paediatric patients to reduce the risk of urine infections and the chance of renal impairment, and also provide urinary continence. All these patients had neurogenic LUT dysfunction bladder as a neonate and/or an adolescent. We have expanded the first paragraph of the Methods to include the age ranges of the group. We have in fact removed the data collected from a three-month control bladder sample, which provided nerve-mediated contractions only, and this has not altered any of the conclusions, and a slight adjustment to these contraction data i.e. Tmax from 7.70 to 7.68 mN.mm-2 and f1/2 from 10.9 to 11.1 Hz. (Results paragraph 1, lines 4 and 8)
We appreciate the neuropathic bladder group was older than the control group due to the different procedures they would undergo and is not something easy to control and yet achieve a sufficient number of biopsy samples. In fact, the age difference between the two groups would tend to reduce the statistical significance of the functional and molecular changes in the two groups due to an increase of contractile function and reduced fibrosis with normal development. We have added a limitation statement, including new ref 36, about this in the Discussion (end of section ‘Histological and biomechanical properties of tissue….’)
4 (Methods, Tissue samples, ethics and preparations) Was the size of the biopsies controlled? The condition of neurogenic bladder is characterized by an increase in the size of the urinary bladder. Did author collect data on bladder sizes? Again, range of ages of the patients involved in this study is big and this translates into differences in whole tissue size, which might impact proteins quantification and comparisons among groups. Please clarify how this was taken into consideration when selecting the tissue for analysis and generating results.
The size of the biopsy samples was limited to what was considered acceptable to the operating surgeon (NJ, PC) and was up to ≈5 mm3 in dimension. These were from the anterior wall of the bladder dome, with none from the trigone or bladder neck. After removal of any adherent mucosa the greater part of the biopsies were used for functional experiments to measure passive stiffness and/or active contractile properties. We dissected out only one preparation for functional experiments from each biopsy sample. The remainder, when available, was used for histology and immunohistochemistry. The size of the biopsy samples was therefore a limitation on the range of experiments we would have liked to perform (see your question 7 below) but preferred to generate fewer but more viable preparations for functional experiments.
Details of biopsy sample size, site and dissection into the preparations are given in the Methods, paragraphs 1 and 3. We have normalised the functional data to preparation size so that the data refer to unit tissue quantities (i.e. tissue cross-section area)
5 (Results, Figure 2) Figure 2 should be shown in the manuscript after figure 1, please correct. Also, the legend should include what the yellow/orange and red staining represents in terms of connective tissue and smooth muscle.
I am not sure what happened about the uploading of the figures into the submitted copy. We will make sure the re-submission order of figures is correct.
The different colour stains have been explained in the expanded figure legend. The explanation of part A is as follows
“A: Verhoeff van Gieson stained sections of tissue from normal (upper trio) and neuropathic (lower trio) bladders. Shown below each primary section (yellow/orange: smooth muscle: red; collagen) are grey scale images of the yellow/orange (left) and red (right) areas obtained from ImageJ files.”
6 (Results, Figure 3) Legend of figure 3 is incomplete. Immunofluorescence images should be accompanied by the description of the markers used. It is not clear how the quantification of the different proteins was performed.
We have expanded the Figure 3 legend and the final sentence of the Multiplex Immunofluorescence section of the Methods. We hope this is now clearer
Quantitation was performed on high magnification images using Huygens software (SVI) [reference 18 - Johal, N et al J Pediat. Urol. 15: 154.e1-154.e9, 2019].
To avoid the clash of colours in the different parts of Figure 3, we have used white and grey bars for the quantification of markers in part B of Figure 3.
7 (Results) In addition to the analysis of proteins associated with fibrosis, it would be relevant to study the innervation of the bladder in the paediatric tissue biopsies. To corroborate the findings in tissue tension/contractility authors should consider conducing immunofluorescence protocols for the detection and quantification of cholinergic and purinergic markers.
It was not possible to carry out the further imaging analysis of measuring nerve fibre number in the detrusor layer due to the limited sizes of the biopsies that were obtained, as they were required for contractility, passive stiffness (functional) properties as well as imaging studies: the functional and the imaging analyses both requiring separate portions of the biopsy. The tissue arrays used for the immunohistochemical studies were completely consumed in preparing sections for the four labels used in the current study so measurement of further epitopes such as muscarinic and purinergic sections was also not possible.
In this context, we have defined a ‘functional denervation’ as a physiological equivalent, by measuring the ratio of force generated by nerve-mediated contractions and carbachol-induced contractions, as the former will generate force via motor nerves and the latter by direct muscle activation. A reduction of the ratio in one group would be defined as a functional denervation. These data are included in the Results section (paragraph 2, lines 5-7 - highlighted) and also included in the Discussion (para 1, lines 5-7 – highlighted).
Reviewer 2 Report
An obvioulsy we--ll-designed and performed expeiment using human tissue samples, investigating functional and structural properties of neuropathic bladders in children compared to controls. Few methodological queries to be addressed:
1. How was the normal bladder function of the control patients established? Were they all submitted to urodynamic investigation? Which urodynamic criteria were applied for neuropaths and controls' recruitment?
2. On what occasion were the bladder samples taken from the neuropathic bladders?
3. How many samples were taken per bladder? How many of them were used for functional experiments and how many for histology and immunohistochemistry? Which areas of the bladder were the samples taken from?
Author Response
Referee 2
Thank you for your careful questions. The responses are below each question, including changes to the text and highlighted in brown.
An obviously well-designed and performed experiment using human tissue samples, investigating functional and structural properties of neuropathic bladders in children compared to controls. Few methodological queries to be addressed:
- How was the normal bladder function of the control patients established? Were they all submitted to urodynamic investigation? Which urodynamic criteria were applied for neuropaths and controls' recruitment?
Details of urodynamic investigations and operative procedures are included in the Methods section (paragraph 1)
- On what occasion were the bladder samples taken from the neuropathic bladders?
Please see answer above, patients were undergoing bladder augmentation.
- How many samples were taken per bladder? How many of them were used for functional experiments and how many for histology and immunohistochemistry? Which areas of the bladder were the samples taken from?
A single biopsy sample was taken from each patient. The size and site of the sample, how the biopsy was prepared for different types of experiment, as well as the size of physiological samples was also raised by referee 1. Responses are in the Methods section, paragraphs 1 and 3 and highlighted in brown and red, as per similar questions from referee 1.
Reviewer 3 Report
The manuscript ‘Functional and structural properties of detrusor from paediatric neuropathic bladders’ aims to detail contractile and biomechanical properties form paediatric population with congenital neurogenic lower urinary tract dysfunction NLUTD as well as to unravel development of (detrusor) fibrosis in these patients. The manuscript reports evaluation of detrusor biopsy samples of patients with congenital NLUTD and also of patients with other not LUTD pediatric patients and compared these.
This is a relevant manuscript and
I advise to scrutinize the terms used: ‘bladder’ should be ‘detrusor’ or ‘lower urinary tract’ in many cases. Also e.g.,: bladder overactivity should be detrusor overactivity etc. Neuropathic bladder should be replaced with neurogenic lower urinary tract dysfunction etc. Reduced compliance should be mentioned as well as reduced sensation of filling (propriocepsis) as well as acontractility.
Congenital development depends also on congenital innervation. As can be observed in lower limb ‘neurogenic’ development: abnormalities in prenatal innervation result in abnormal (legs) morphology, in abnormal development of muscles (and consequently of bone tissue) and abnormal (and in-) ability of muscle to contract and relax as well as abnormal (and lack of) sensation and awareness and lack of cerebral (mental) projection of regions without sensation. Similar to lower limb abnormality; development of lower urinary tract, including bladder and pelvic floor muscles is prone to result in abnormal morphology as well as in sensory and motoric (and bodily awareness) functions. Macroscopic morphology may be reflected in microscopic morphology and alterations in relevant proteins. Abnormal morphology and (dys-) function is highly variable in patients with MMC (and –as an aside- clinically not related to limb abnormalities in a given person.) The MMC patients where all >2years (or very much older) when tissue was harvested and congenital abnormalities may have been developed into secondary (iatrogenic) abnormalities and inflammation. The reason for the chance to harvest tissue in these MMC patients (during surgery) was not given. Information on the (range of) dysfunctions is lacking. This all reduces the generalizability of the observations and ‘sub’-conclusions as: The SM content of neuropathic tissue sections was less than that of normal bladders’ is (as an example) like an unsupported generalization. I request to consider correction of this type of statements where necessary. And: is ref 3 a good reference to show that ‘prevalence remains high despite….’would a reference that report incidence over time (decades) –with key events of understanding- not be more relevant?
Author Response
Referee 3
Thank you for your valuable questions. The responses are below each question, including changes to the text and highlighted in blue.
The manuscript ‘Functional and structural properties of detrusor from paediatric neuropathic bladders’ aims to detail contractile and biomechanical properties form paediatric population with congenital neurogenic lower urinary tract dysfunction NLUTD as well as to unravel development of (detrusor) fibrosis in these patients. The manuscript reports evaluation of detrusor biopsy samples of patients with congenital NLUTD and also of patients with other not LUTD pediatric patients and compared these.
This is a relevant manuscript and
1 I advise to scrutinize the terms used: ‘bladder’ should be ‘detrusor’ or ‘lower urinary tract’ in many cases. Also e.g.,: bladder overactivity should be detrusor overactivity etc. Neuropathic bladder should be replaced with neurogenic lower urinary tract dysfunction etc. Reduced compliance should be mentioned as well as reduced sensation of filling (propriocepsis) as well as acontractility.
We have gone through the manuscript and marked changes as suggested. In particular, the appropriate use of ‘detrusor’ and ‘detrusor over(under)activity. We have also changed
We have also changed the manuscript title, subject to your approval
“Neurogenic lower urinary tract dysfunction and changes to detrusor functional properties”
References [6,7] to low bladder compliance and reduced sensation on filling in some patients with neurogenic LUT dysfunction, as well as EAU/ESPU guidelines recommending urodynamics have been added.
2 Congenital development depends also on congenital innervation. As can be observed in lower limb ‘neurogenic’ development: abnormalities in prenatal innervation result in abnormal (legs) morphology, in abnormal development of muscles (and consequently of bone tissue) and abnormal (and in-) ability of muscle to contract and relax as well as abnormal (and lack of) sensation and awareness and lack of cerebral (mental) projection of regions without sensation. Similar to lower limb abnormality; development of lower urinary tract, including bladder and pelvic floor muscles is prone to result in abnormal morphology as well as in sensory and motoric (and bodily awareness) functions. Macroscopic morphology may be reflected in microscopic morphology and alterations in relevant proteins. Abnormal morphology and (dys-) function is highly variable in patients with MMC (and – as an aside- clinically not related to limb abnormalities in a given person.) The MMC patients where all >2years (or very much older) when tissue was harvested and congenital abnormalities may have been developed into secondary (iatrogenic) abnormalities and inflammation. The reason for the chance to harvest tissue in these MMC patients (during surgery) was not given. Information on the (range of) dysfunctions is lacking.
Children with neuropathic bladder may require surgery at any time to reduce the risk of urine infection and to protect the upper tracts and renal function. Some surgeries such as a vesicostomy may be required in infancy and others such as bladder augmentation could be performed at any age to adulthood, as with these patients.
We have focussed on a study of the pathophysiology of neurogenic LUT dysfunction in paediatric patients. We appreciate that these may be secondary changes to the initial causal event or will have arisen directly. The data give insight into the cellular and tissue changes that may have occurred and do not offer a reason why they might have occurred
3 This all reduces the generalizability of the observations and ‘sub’-conclusions as: The SM content of neuropathic tissue sections was less than that of normal bladders’ is (as an example) like an unsupported generalization. I request to consider correction of this type of statements where necessary.
We appreciate your point. We have included a paragraph at the very beginning of the Discussion to address the issue of how generalised will be the conclusions to be drawn from this study. This was also raised in a similar guise by referee 1
4 is ref 3 a good reference to show that ‘prevalence remains high despite….’would a reference that report incidence over time (decades) –with key events of understanding- not be more relevant?
We have replaced the reference with another that had a primary objective to compare pre- and post-natal repair of myelomingocoele. However, it included data. in a first-world country. on its prevalence (up to 2011) and the stabilisation of its prevalence
Round 2
Reviewer 1 Report
Authors' answers to reviwers were clear and objective and the manuscript was notably improved in accordance with the comments.
Author Response
Thank you for your comments which has helped to to improve the manuscript
Reviewer 3 Report
The manuscript ‘Functional and structural properties of detrusor from paediatric neuropathic bladders.’ was resubmitted with changes made, based on the reviewers comments. The title has been changed to ‘Neurogenic lower urinary tract dysfunction and changes to detrusor functional properties.’ Some adjustments have been made to the use of words, which I think is good. However, my problem of generalizability is by no means solved. Not even in the authors' answers. I read in the conclusion: ‘Detrusor from children with neurogenic LUT dysfunction is less contractile, compared to tissue from normally-functioning bladders and is associated with loss of smooth muscle compared to connective tissue.’ Which is ‘summarized’ in the abstract: ‘We conclude paediatric neuropathic bladders have poor contractility associated with reduced smooth muscle. The increased stiffness would manifest as reduced filling compliance that may contribute to raised upper tract pressures and renal dysfunction.’ These conclusions are much too generalized and also the second sentence of the conclusion of the abstract is not based on the current study. (I think this is more the argument to do the lab study). First: the tissue was harvested from children with meningomyelocele; a congenital abnormality of the neural tube development (that may have its specific ‘wnt-problems’) that results in dysfunction secondary to this congenital abnormality. Secondary, the comparator group has –likely- had vesicoureteral reflux, (or open urachus); maybe also congenital (WNT?) abnormalities resulting in weak interstitium, or to weak muscles may have originated these pathologies. These (comparator) bladders/detrusors are not certainly 'normal'. Third: children born with MMC are prone to have medical management for their dysfunction. Medication, oral or injected (affecting cell receptors?), catheters and may have more frequent or especially more chronical urinary tract infections, a subgroup (especially those with stiff and small bladders!) gradually evolve to surgical management to increase bladder capacity and reduce pressure and achieve continence. Sometimes after earlier surgical interventions. Not only age is the confounder in this study, not only age makes that the differences observed are (not) ‘congenital’. The words ‘change(s)’; ‘decline’; and ‘reduction’ throughout the manuscript need careful reconsideration. And any reference to the cause of the differences between these cohorts also.
However, as a simple clinician, I am also asking for a clarification that may not be necessary for most readers of this journal (or article). I think the WNT signaling pathway is fixed from conception, but I'm not sure if signaling molecules can change afterwards. That's why I'm not sure if the last sentence in the abstract can stay that way.
Finally, I don't think the new title is an improvement. That has to do with all my current and previous comments.
Author Response
Referee 3
The manuscript ‘Functional and structural properties of detrusor from paediatric neuropathic bladders.’ was resubmitted with changes made, based on the reviewers comments. The title has been changed to ‘Neurogenic lower urinary tract dysfunction and changes to detrusor functional properties.’ Some adjustments have been made to the use of words, which I think is good.
However, my problem of generalizability is by no means solved. Not even in the authors' answers. I read in the conclusion: ‘Detrusor from children with neurogenic LUT dysfunction is less contractile, compared to tissue from normally-functioning bladders and is associated with loss of smooth muscle compared to connective tissue.’ Which is ‘summarized’ in the abstract: ‘We conclude paediatric neuropathic bladders have poor contractility associated with reduced smooth muscle. The increased stiffness would manifest as reduced filling compliance that may contribute to raised upper tract pressures and renal dysfunction.’ These conclusions are much too generalized and also the second sentence of the conclusion of the abstract is not based on the current study. (I think this is more the argument to do the lab study).
First: the tissue was harvested from children with meningomyelocele; a congenital abnormality of the neural tube development (that may have its specific ‘wnt-problems’) that results in dysfunction secondary to this congenital abnormality.
Secondary, the comparator group has –likely- had vesicoureteral reflux, (or open urachus); maybe also congenital (WNT?) abnormalities resulting in weak interstitium, or to weak muscles may have originated these pathologies. These (comparator) bladders/detrusors are not certainly 'normal'.
Third: children born with MMC are prone to have medical management for their dysfunction. Medication, oral or injected (affecting cell receptors?), catheters and may have more frequent or especially more chronical urinary tract infections, a subgroup (especially those with stiff and small bladders!) gradually evolve to surgical management to increase bladder capacity and reduce pressure and achieve continence. Sometimes after earlier surgical interventions.
Not only age is the confounder in this study, not only age makes that the differences observed are (not) ‘congenital’.
The words ‘change(s)’; ‘decline’; and ‘reduction’ throughout the manuscript need careful reconsideration. And any reference to the cause of the differences between these cohorts also.
However, as a simple clinician, I am also asking for a clarification that may not be necessary for most readers of this journal (or article). I think the WNT signaling pathway is fixed from conception, but I'm not sure if signaling molecules can change afterwards. That's why I'm not sure if the last sentence in the abstract can stay that way.
Finally, I don't think the new title is an improvement. That has to do with all my current and previous comments.
Authors’ reply
We fully appreciate your comments and you have raised an important aspect that is relevant to any research that uses human tissue samples, but is one that is less commonly discussed. The key aspect, as we see it, is a comparison of the properties of detrusor muscle from children who had a developmental spinal abnormality and consequent LUT dysfunction with tissue from children who have no such dysfunction. Our central theme of this work is to determine if fibrosis is an important contributor to urinary tract dysfunction. Both groups in this study will represent heterogenous populations, but the simple question was if there were overall differences of structure and function despite this heterogeneity.
We have made the following changes.
1 Abstract In the final three sentences of the Abstract we have tried to keep the conclusion within the scope of the study and not extrapolate to clinical sequelae.
2 Methods We have defined the two groups as a neuropathic group and a control a (comparator) group (lines 2 and 5 of the 1st paragraph). And used these terms throughout
3 Discussion The first paragraph is our attempt to address the issues you raise in listing the confounders that may contribute to data variability within a group. We would argue that despite cohort heterogeneity structure-function differences may still be observed.
Thus, we have presented the data as an observational study. We have defined the two study groups as those with and without lower urinary tract dysfunction, without allusion to a comparator group representing true normality. We take your point about certain comparative words such as ‘changes’ as representing a progressive pathway from normality and have tried to merely state the fact that values differ between the two groups. We hope this is sufficient.